# An observational cohort study comparing ibuprofen and oxycodone in children with fractures

Samina Ali[1]*, Robin Manaloor[2], David W. Johnson[3], Rhonda J. Rosychuk[4], Sylvie LeMay[5], Bruce Carleton[6], Patrick J. McGrath[7], Amy L. Drendel[8], on behalf of Pediatric Emergency Research Canada[¶]

1 Faculty of Medicine & Dentistry and Women & Children's Health Research Institute (WCHRI), Department of Pediatrics, University of Alberta, Edmonton, Alberta, Canada, 2 Faculty of Medicine & Dentistry, Edmonton, AB, Canada, 3 Department of Pediatrics, Alberta Children's Hospital, University of Calgary, Calgary, Alberta, Canada, 4 Faculty of Medicine & Dentistry, Department of Pediatrics, University of Alberta, Edmonton, Alberta, Canada, 5 Faculty of Nursing, Université de Montréal, Montreal, QC, Canada, 6 Division of Translational Therapeutics, Department of Pediatrics, University of British Columbia, Vancouver, British Columbia, Canada, 7 Department of Psychology, IWK Health Centre, Dalhousie University, Halifax, Nova Scotia, Canada, 8 Department of Pediatrics, Medical College of Wisconsin, Milwaukee, WI, United States of America

¶ Membership of Pediatric Emergency Research Canada is provided in the Acknowledgments
* sali@ualberta.ca

**Data Availability Statement:** Data cannot be shared publicly because of consent and confidentiality reasons. Data are available from the Research Informatics Lead, Women and Children's

## Abstract

### Objective

To compare the effectiveness and safety of prescribing ibuprofen and oxycodone for at-home management of children's fracture pain.

### Methods

A prospective observational cohort was conducted at the Stollery Children's Hospital pediatric emergency department (June 2010-July 2014). Children aged 4–16 years with an isolated fracture discharged home with advice to use either ibuprofen *or* oxycodone were recruited.

### Results

A cohort of 329 children (n = 217 ibuprofen, n = 112 oxycodone) were included. Mean age was 11.1 years (SD 3.5); 68% (223/329) were male. Fracture distribution included 80.5% (264/329) upper limb with 34.3% (113/329) requiring fracture reduction. The mean reduction in Faces Pain Score-Revised score (maximum pain–post-treatment pain) for Day 1 was 3.6 (SD 1.9) (ibuprofen) and 3.8 (SD 2.1) (oxycodone) (p = 0.50); Day 2 was 3.6 (SD 1.8) (ibuprofen) and 3.7 (SD 1.6) (oxycodone) (p = 0.56); Day 3 was 3.7 (SD 1.7) (ibuprofen) and 3.3 (SD 1.7) (oxycodone) (p = 0.24). Children prescribed ibuprofen (51.2%, 109/213) experienced less adverse events compared to those prescribed oxycodone (70.5% 79/112) on Day 1 (p = 0.001). Children prescribed ibuprofen (71.8%, 150/209) had their function (eat, play, school, sleep) affected less than those prescribed oxycodone (83.0%, 93/112) (p = 0.03) on Day 1.

Health Research Institute Mr. Rick Watts (rick.
watts@ualberta.ca) or the corresponding author
Dr. Samina Ali (sali@ualberta.ca) for researchers
who meet the criteria for access to confidential
data.

**Funding:** This study was funded by the Canadian
Institutes of Health Research (CIHR-DSEN (2010-
2011) FRN 103534 and CIHR-DSEN (2012-2013)
FRN 120529 for which Dr. Samina Ali was the
principal investigator. The CIHR did not have any
role in the collection, management, analysis or
interpretation of data; writing of the report; or the
decision to submit for publication.

**Competing interests:** The authors have declared
that no competing interests exist.

## Conclusion

Children prescribed ibuprofen or oxycodone experienced similar analgesic effectiveness for at-home fracture pain. Oxycodone prescribing was associated with more adverse events and negatively impacted function. Oxycodone use does not appear to confer any benefit over ibuprofen for pain relief and has a negative adverse effect profile. Ibuprofen appears to be a safe option for fracture-related pain.

## Introduction

Pain is among the most common reasons for seeking healthcare and choosing appropriate pain medication is critical [1–3]. The American Academy of Pediatrics and the World Health Organization have mandated that children's pain be addressed as a fundamental human right [4,5] and sub-optimal treatment is no longer accepted by patients, families, or healthcare providers [6–8]. Children's fracture pain is under-treated both in the hospital and in the at-home setting [9–12]. Non-steroidal anti-inflammatory drugs (NSAIDs) alone, or in combination with oral opioids, are the most commonly prescribed analgesics for fracture pain [13–15]. While ibuprofen has emerged as the current first-line medication for this type of pain [15], it has also been shown to be inadequate for those with more significant pain [16]. Opioids are commonly used for moderate to severe traumatic pain in children, but this practice has come under recent scrutiny, in light of the current opioid crisis and concerns of misuse potential after short-term use [17–20]. While oral morphine has been shown to have similar effectiveness as oral ibuprofen for fracture-related pain management [21–24], oxycodone has not been studied to the same extent. As such, limited evidence exists to inform an evidence-based approach to management of fracture pain in children. Our study objectives were to describe the at-home clinical effectiveness and safety of ibuprofen and oxycodone when they are recommended for use and determine effects on short-term functioning for children discharged home from the emergency department (ED) with a limb fracture.

## Materials and methods

### Study design and setting

This study was a prospective observational cohort study conducted at the Stollery Children's Hospital (Edmonton, Alberta, Canada). The Stollery Children's Hospital is a tertiary care facility with 155 beds that receives children from north and central Alberta, as well as parts of Saskatchewan, Manitoba, British Columbia, the Northwest Territories, Yukon, and Nunavut [25]. Mean annual census for the emergency department at the time of the study (2010–2014) was 36,000. This study was approved by the University of Alberta's Health Research Ethics Board (Pro00005942).

### Participants

Patient recruitment occurred from June 2010 to July 2014. Children were eligible for the study if they presented with an isolated, acute limb fracture (<24 hours old), were between 4 to 16 years and were discharged home with *either* ibuprofen or oxycodone as the recommended choice for pain treatment medication. Exclusion criteria included (a) utilizing daily chronic pain medications, (b) being prescribed ibuprofen *and* oxycodone upon discharge, (c) cognitively unable to self-report pain, (d) unable to communicate in English and (e) no telephone

access for follow-up. Both prescribing of the medications and instructions on how to use them at home were left to the clinical discretion of the prescribing physician, as this was an observational (and not interventional) study.

## Recruitment

All participants provided written informed consent (caregiver) and assent (child) before participation. A trained research assistant conducted eligibility assessments, collected basic demographic data from the families, and conducted a medical record review. Upon discharge, participants were provided a copy of the Faces Pain Scale–Revised (FPS-R) [26] for reference, and a medication and symptom logbook. Caregivers were called once per day for three days, reflecting the period of maximal pain following acute injury [27]. During each call, the caregiver was asked to report their child's **self-reported** maximal, minimal, and post-treatment (if applicable) pain scores for the given day, which they had *collected directly* from their child. Questions were also asked regarding medication use, and adverse events. Specifically, questions regarding whether fracture pain affected their child's quality of life (QoL) included inquiry regarding appetite, sleep, school attendance, and play. These QoL measures were chosen based on previously published literature regarding post-fracture functional impairment [28]. Follow up phone calls were made at two and six weeks post-injury. At this time, information regarding any changes to final diagnosis or management, progression of healing and follow up appointments was obtained. (See S1 File).

## Data collection and management

Data regarding age, sex, fracture location, pain measurement, adverse events and medication usage were collected via direct interview at the time of recruitment, chart review, and through follow up phone calls. A data entry specialist performed single data entry directly into Open-Clinica [29]. OpenClinica case report forms were configured to validate the data at the point of entry (e.g. patient age entered must be <18), keeping data entry errors to a minimum. Data coding for adverse events was recorded using the Medical Dictionary for Regulatory Activities (MedDRA), a standardized terminology for reporting adverse events [30].

## Measures

The primary outcome was pain score reduction as measured by the FPS-R on Day 1 after ED discharge. Secondary outcomes were a) presence of at least one adverse event on Day 1 after ED discharge, b) pain score reduction on Days 2 and 3, c) presence of at least one adverse event on Days 2 and 3, and d) effects on quality of life for children. The FPS-R is a 6-item self-report measure to assess intensity of pain. It has been validated for use in children 4 years of age and older [26]. The scale is scored from 0 (no pain) to 10 (maximum amount of pain) by choosing the appropriate face on a horizontal axis. QoL measures were scored as binary (present/absent). Adverse events were also scored as binary (present/absent). Outcomes and measurements were formulated prior to data collection.

## Sample size and power

The study required 50 children in both groups defined by the primary outcome to be able to detect an effect size of 0.5 (using the FPS-R) for comparing means with 80% power and a one-sided ($\alpha = 0.05$), two-sample, t-test. Tests were planned for the ibuprofen and oxycodone medication groups separately, requiring a total of 100 children. Even with the analyses for the

secondary outcomes based on two-sided tests, the increased sample size meant that we could detect effect sizes smaller than 0.5 between the two medication groups.

## Statistical analysis

For each medication group, univariable summaries (means, medians, standard deviations [SDs], ranges) were provided for continuous variables (e.g., time of presentation, FPS-R score), and frequency distributions summarized categorical variables (e.g., sex). Two-sided, two-sample, t-tests compared mean pain scores for patients in each medication group. Chi-squared tests (or Fisher's Exact Tests for small cell counts) compared medication group and categorical variables. Analyses were conducted at each time point separately (i.e., Day 1, Day 2, Day 3, Week 2, and Week 6). Chi-squared tests were used to compare the proportion of children who experienced at least one adverse event during the first 3 days post-injury. The statistical package R [31] was used and a p-value $< 0.05$ was considered statistically significant.

## Results

### Demographic characteristics

For this study, 2539 children were screened of which 2210 were excluded; top reasons for exclusion were a) did not receive a prescription for the drug of interest, b) fracture was greater than 24 hours old, and c) the patient was admitted. A total of 329 children (n = 217 ibuprofen, n = 112 oxycodone) were included (Table 1).

### Pain medication use

Fig 1 reports the number of doses of ibuprofen or oxycodone used, by group. Mean dose of administered ibuprofen was 9.2 mg/kg/dose (SD 3.5) on Day 1, 9.0 mg/kg (SD 3.8) on Day 2, and 8.5 mg/kg (SD 3.8) on Day 3; these are all within the recommended dosing range of 5-10mg/kg/dose. Mean dose of administered oxycodone was 0.11 mg/kg/dose (SD 0.05) on Day 1, 0.10 mg/kg (SD 0.04) on Day 2, and 0.09 mg/kg (SD 0.04) on Day 3; these are all within the recommended dosing range of 0.05–0.15 mg/kg/dose. On Day 1, a total of 19% (41/213) of the ibuprofen group used other medication(s) for pain treatment compared to 42% (47/112) in the oxycodone group (p $<$ 0.001). On Day 2, a total of 18% (37/212) of the ibuprofen group used other medication(s) for pain treatment compared to 48% (52/108) in the oxycodone group (p $<$ 0.001). On Day 3, a total of 12% (25/209) of the ibuprofen group used other medication (s) for pain treatment compared to 46% (47/103) in the oxycodone group (p $<$ 0.001). See Table 2 for medications used.

### Pain measures

Table 3 provides a summary of mean pain scores and pain reduction from Days 1–3. On Day 1, 90% (293/325) of children reported pain, with 88% (188/213) in the ibuprofen group and 94% (105/112) in the oxycodone group (p = 0.17). The mean reduction in pain score (maximum pain–post-treatment pain) as measured by the FPS-R for Day 1 was 3.6 (SD 1.9) (ibuprofen) and 3.8 (SD 2.1) (oxycodone) (p = 0.50). On Day 2, 83% (267/320) of children reported pain, with 80% (169/212) in the ibuprofen group and 91% (98/108) in the oxycodone group (p = 0.019). The mean reduction in pain score on Day 2 was 3.6 (SD 1.8) (ibuprofen) and 3.7 (SD 1.6) (oxycodone) (p = 0.56). On Day 3, 69% (214/312) of children reported pain, with 64% (133/209) in the ibuprofen group and 79% (81/103) in the oxycodone group (p = 0.011). The mean reduction in pain score for Day 3 was 3.7 (SD 1.7) (ibuprofen) and 3.3 (SD 1.7) (oxycodone) (p = 0.24).

**Table 1. Demographics characteristics (n = 329).**

| | Ibuprofen Group | Oxycodone Group | Total Cohort |
|---|---|---|---|
| Age, mean (SD), years | 10.4 (3.7) | 12.3 (2.9) | 11.0 (3.5) |
| Weight, mean (SD), kg | 42.6 (19.0) | 48.9 (15.6) | 44.7 (18.1) |
| Sex (Boys) n (%) | 136 (62.7) | 87 (77.7) | 223 (67.8) |
| Fracture Location n (%) | | | |
| Forearm/Wrist | 106 (48.8) | 48 (43.2) | 154 (47.0) |
| Upper Arm/Elbow | 30 (13.8) | 9 (8.1) | 39 (11.9) |
| Shoulder/Clavicle | 14 (6.5) | 28 (25.2) | 42 (12.8) |
| Hand | 23 (10.6) | 6 (5.4) | 29 (8.8) |
| Lower Leg/Ankle | 34 (15.7) | 12 (10.8) | 46 (14.0) |
| Upper Leg/Hip | 2 (0.9) | 4 (3.6) | 6 (1.8) |
| Foot | 8 (3.7) | 4 (3.6) | 12 (3.6) |
| Procedural Sedation n (%) | 64 (29.5) | 41 (36.6) | 105 (31.9) |
| Fracture reduction performed n (%) | 70 (32.3) | 43 (38.4) | 113 (34.3) |
| Buckle Fracture n (%) | 24 (11.1) | 5 (4.5) | 29 (8.8) |
| Post-ED Discharge Follow Up n (%) | | | |
| Not charted/Unable to ask MD | 22 (10.1) | 2 (1.8) | 24 (7.3) |
| Return to ED, PRN | 3 (1.4) | 1 (0.89) | 4 (1.2) |
| Return to ED, Scheduled | 4 (1.8) | 0 (0.0) | 4 (1.2) |
| Referral to orthopedic surgeon | 145 (67.7) | 87 (77.7) | 234 (71.1) |
| F/U with family doctor | 30 (13.8) | 10 (8.9) | 40 (12.2) |
| Other F/U* | 13 (6.5) | 12 (5.5) | 26 (7.9) |

*Referral to other specialty (plastic surgery, vascular surgery); sports medicine clinic; radiology for further imaging.

## Adverse events

Table 4 presents cumulative adverse events by study group and type. On Day 1, 58% (188/325) of all children reported adverse events, with 51% (109/213) in the ibuprofen group and 71% (79/112) in the oxycodone group (p = 0.001). On Day 2, 43% (138/319) of all children reported adverse events, with 37% (78/211) in the ibuprofen group and 56% (60/108) in the oxycodone group (p = 0.002). On Day 3, 28% (88/312) of all children reported adverse events, with 24% (50/209) in the ibuprofen group and 37% (38/103) in the oxycodone group (p = 0.024).

## Functional outcomes

Fig 2A–2C presents function affected by group. Fig 3 presents functional outcomes by pain severity.

## Week 2 follow up

At Week 2, 23.6% (47/199) still reported pain in the ibuprofen group and 26.9% (28/104) in the oxycodone group (p = 0.622). Mean pain for ibuprofen was 3.96 (SD 2.21) and for oxycodone was 3.27 (SD 1.86) (p = 0.152). Function affected in the ibuprofen group was 24.9% (49/197) including ability to eat normally [2.0% (4/197)]; play [21.3% (42/197)]; school [6.7% (13/197)] and sleep [7.1% (14/197)]; function affected in the oxycodone group was 32.7% (34/104) including ability to eat normally [3.8% (4/104)]; play [28.8% (30/104)]; school [7.7% (8/104)] and sleep [10.6% (11/104)] (p = 0.191).

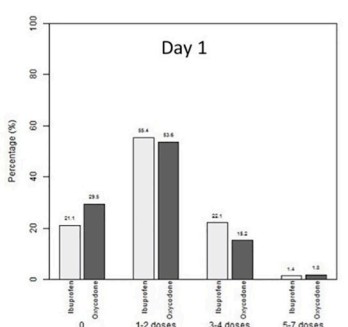
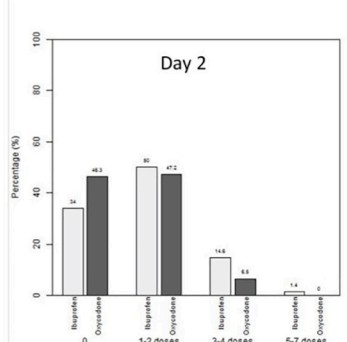
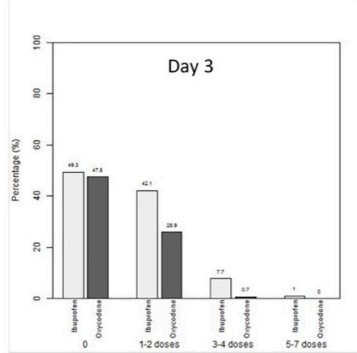

**Fig 1. Number of doses of recommended medication used per day (Days 1–3).**

## Week 6 follow up

At Week 6, 12.6% (25/199) still reported pain in the ibuprofen group and 6.7% (7/104) in the oxycodone group (p = 0.170). Mean pain for ibuprofen was 3.00 (SD 2.07) and for oxycodone was 2.36 (SD 2.34) (p = 0.410). Function affected in the ibuprofen group was 11.0% (22/200) including ability to eat normally [2.0% (4/197)]; play [10.2% (20/197)]; school [2.5% (5/197)] and sleep [2.5% (5/197)]; function affected in the oxycodone group was 4.8% (5/104) including ability to eat normally [0.96% (1/104)]; play [4.8% (5/104)]; school [0.96% (1/104)] and sleep [1.9% (2/104)] (p = 0.112).

**Table 2. At-home pain management.**

|  | Ibuprofen Group (n = 213) n (%) |  |  | Oxycodone Group (n = 112) n (%) |  |  |
|---|---|---|---|---|---|---|
| **Drug** | **Day 1** | **Day 2** | **Day 3** | **Day 1** | **Day 2** | **Day 3** |
| **No medication used** | 39 (18.3) | 38 (17.8) | 69 (32.4) | 9 (8.0) | 14 (12.5) | 25 (22.3) |
| **Ibuprofen** | **168 (78.9)** | **140 (65.7)** | **106 (49.8)** | 31 (27.7) | 29 (25.9) | 29 (25.9) |
| **Oxycodone*** | 1 (0.5) | 0 (0.0) | 1(0.5) | **79 (70.5)** | **58 (51.8)** | **37 (33.0)** |
| **Acetaminophen** | 36 (16.9) | 32 | 21 | 15 (13.4) | 22 (19.6) | 18 (16.1) |
| **Codeine*** | 3 (1.4) | 2 (0.9) | 2 (0.9) | 1 (0.9) | 1 (0.9) | 0 (0.0) |
| **Naproxen** | 1 (0.5) | 2 (0.9) | 1 (0.5) | 0 (0.0) | 0 (0.0) | 0 (0.0) |
| **Tramacet** | 1 (0.5) | 1 (0.5) | 1(0.5) | 0 (0.0) | 0 (0.0) | 0 (0.0) |

*Alone or in Combination with Acetaminophen as a marketed combination pill; please note multiple medication options may have been used at the same time, allowing for the column total to be >100%.

**Table 3. Mean pain scores, Days 1–3.**

| Pain Score | Day 1 | | Day 2 | | Day 3 | |
|---|---|---|---|---|---|---|
| | Ibuprofen (n = 190) | Oxycodone (n = 108) | Ibuprofen (n = 179) | Oxycodone (n = 103) | Ibuprofen (n = 152) | Oxycodone (n = 89) |
| Delta Pain (max–post treatment) | 3.6 (1.9) | 3.8 (2.1) | 3.6 (1.8) | 3.7 (1.6) | 3.7 (1.7) | 3.3 (1.7) |
| Maximum Pain | 6.0 (2.4) | 6.3 (2.3) | 5.3 (2.4) | 5.7 (2.3) | 4.4(2.6) | 4.9 (2.4) |
| Minimum Pain | 2.0 (1.9) | 2.4 (1.9) | 1.6 (1.7) | 1.8 (1.7) | 1.2 (1.7) | 1.4 (1.6) |
| Average Daily Pain | 4.2 (2.1) | 4.4 (2.4) | 3.6 (2.0) | 3.8 (2.2) | 3.0 (2.1) | 3.1 (2.0) |
| Post-Treatment Pain | 2.4 (2.1) | 2.7 (2.1) | 2.2 (2.0) | 2.3 (1.9) | 1.8 (1.9) | 1.7 (1.9) |

## Discussion

Children with fractures experience pain and functional limitations at home. The current study demonstrated that children reported moderate pain during the first three days after fracture, and that pain medications were used with varying daily frequency. No differences were seen in maximum and minimum pain scores between groups. More children in this observational cohort study had ibuprofen recommended for at-home pain treatment than oxycodone, which is in keeping with current clinical practice [10]. Children in both ibuprofen and oxycodone groups experienced reduced maximum pain to the mild/moderate range with comparable analgesic effectiveness. Adverse events were reported in approximately two-thirds of children; notably, those in the ibuprofen group experienced significantly less adverse events compared to the oxycodone group. Functional outcomes (i.e., appetite, play, school, sleep) were significantly more negatively affected in the oxycodone group compared to the ibuprofen group on the first and second days after discharge.

This cohort study's findings regarding the clinical effectiveness of prescribing ibuprofen and oxycodone are similar to an ED trial comparing ibuprofen, oxycodone, and the combination of both medications which demonstrated no significant differences among the groups in terms of analgesia [32]. It also mirrors results of trials that have looked at other oral opioids for orthopedic injury and minor surgery-related pain. A pediatric study that compared ibuprofen versus oral morphine for post-operative pain management after minor orthopedic surgery found that both drugs decreased pain similarly; however, oral morphine resulted in greater adverse events [22]. Another post-surgical study followed outpatients after discharge and found no significant differences in pain scores between acetaminophen/codeine versus acetaminophen/ibuprofen and also demonstrated that the use of acetaminophen/codeine resulted

**Table 4. Cumulative adverse events (Days 1–3) by study group.**

| | Ibuprofen Group | Oxycodone Group | Total N (%) | p Value |
|---|---|---|---|---|
| **Adverse Event** | | | | |
| **All** | 129/213 (60.6) | 91/112 (81.2) | 220/325 (67.7) | **<0.001** |
| **Abdominal Pain** | 20/209 (9.6) | 20/103 (19.4) | 40/312 (12.8) | **0.0144** |
| **Appetite Loss** | 69/212 (32.5) | 44/102 (43.1) | 113/314 (36.0) | 0.0671 |
| **Constipation** | 23/206 (11.2) | 19/100 (19.0) | 42/306 (13.7) | 0.0617 |
| **Dizziness** | 20/208 (9.6) | 37/104 (35.6) | 57/312 (18.3) | **<0.001** |
| **Drowsiness** | 90/210 (42.9) | 72/108 (66.7) | 162/318 (50.9) | **<0.001** |
| **Nausea** | 31/208 (14.9) | 39/103 (37.9) | 70/311 (22.5) | **<0.001** |
| **Rash** | 8/208 (3.8) | 6/101 (5.9) | 14/309 (4.5) | 0.4063 |
| **Vomiting** | 6/208 (2.9) | 16/102 (15.7) | 22/310 (7.1) | **<0.001** |

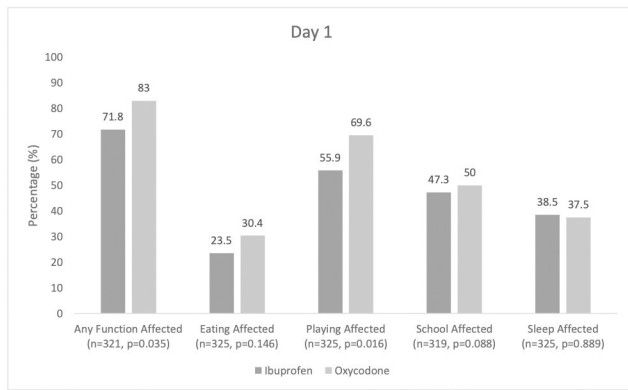
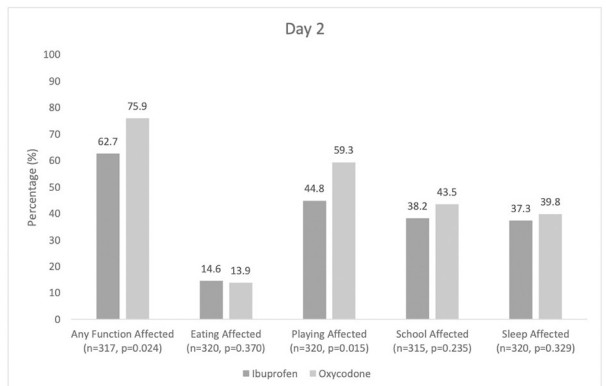

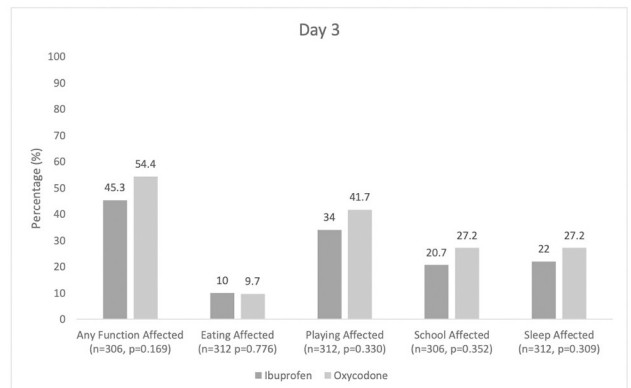

**Fig 2. A.** Functional outcomes by group (Day 1). **B.** Functional outcomes by group (Day 2). **C.** Functional outcomes by group (Day 3).

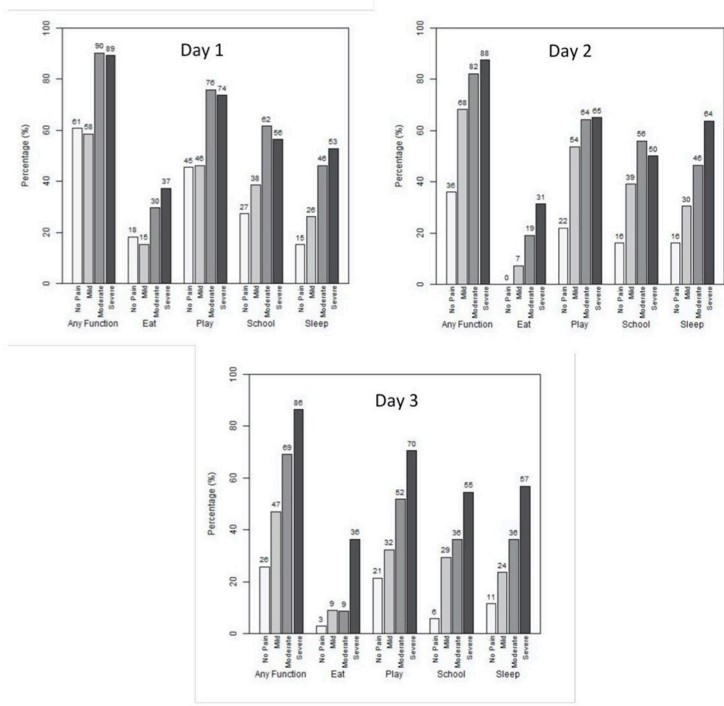

**Fig 3. Function affected vs. pain severity (n = 329).**

in more adverse events and higher discontinuation rates [33]. Lastly, a randomized trial comparing outpatient ibuprofen versus acetaminophen with codeine use in children with arm fractures showed no significant differences in pain reduction between the medication groups; notably, the proportion of children who had any function affect by pain was lower in the ibuprofen group along with lower reported adverse events [28]. Health Canada and FDA warnings [34,35] from recent years have now steered clinicians away from codeine, and despite these early studies, we have yet to find an appropriate oral opioid for orthopedic injury-related pain.

Among children with acute fracture pain, the presented results support clinical trial evidence that oxycodone has no analgesic benefit over ibuprofen for first-line treatment of moderate to severe orthopedic-injury related pain [32,33]. To our knowledge, this is the one of few studies to compare analgesic effectiveness between ibuprofen and oxycodone, and the only one to study its effects *after* discharge from the ED. The after discharge, post-market analysis of these medications is essential to understanding the true effectiveness of these medications and represents a needed complement to controlled clinical trials. Notably, in this study, over 40% of the oxycodone group used alternative non-opioid medication on the first day after discharge; this information would potentially not have been captured in a clinical trial, if limitations were placed on the use on non-assigned trial medications. It is important to note that when one considers that there may be some inherent clinical or demographic differences in the participants included in each of the cohorts, coupled with the use of other pain medications besides their original assignment, our study results do not necessarily reflect the 'pure' clinical efficacy of oxycodone, but rather, the clinical effectiveness of prescribing it for fracture-related pain.

While pain score reduction was not statistically different between ibuprofen and oxycodone groups, important differences were observed for children in terms of adverse events. In the current study, those prescribed oxycodone had a significantly worse adverse event profile compared to children prescribed ibuprofen, with over 80% of children in the oxycodone group reporting cumulative adverse events in the first three days after discharge compared to 60% in the ibuprofen group. The report of adverse events in the current study was comparable to that of other orthopedic injury-related analgesia trials in the outpatient setting [22,32,33]. This is also consistent with a systematic review that concluded that ibuprofen and acetaminophen have similar reported adverse event profiles, and notably less adverse events than opioids [24]. A recent randomized trial comparing oral morphine versus ibuprofen or both in combination in children with orthopedic injury resulted in significantly more adverse events in the morphine arms (21%) than those in the ibuprofen only arm (6.6%) [13]. The most common adverse events seen in the morphine arms were nausea, abdominal pain and drowsiness. Similarly, in the current study, one of the most commonly reported adverse events was drowsiness experienced in two-thirds of the oxycodone group and just under half of the ibuprofen group. This is in keeping with previous studies, as well [22,23].

While pain associated with fractures has been well described and studied, related functional impairment is less well understood [28]. Just over three-quarters of children with fractures in our study experienced decreased functional activity in at least one domain, which included appetite, play, school and sleep. This is similar to a previous study which described 80% of children with compromise in at least one domain [27]. Function is a critical and patient-focused indicator of good fracture pain management and correlates with child and parent satisfaction [9]. Despite this, it remains under-utilized in research and literature. Improving functional outcomes is a child-oriented outcome measure that provides a more holistic approach to pediatric fracture pain management. For every day of persistent pain for a child, there are associated decreases in health-related quality of life and family disruptions [36]. Measuring and

improving such outcomes may avoid the undertreatment of pain and functional limitations along with associated negative consequences [37].

There were several potential limitations to the current study. The study design was observational in nature and not a clinical trial. As such, direct causal relationships cannot be confirmed regarding therapeutic effectiveness and safety. Nonetheless, the quality of data and methods applied may allow for meaningful preliminary conclusions to be drawn. Second, parents administered analgesic medication at their discretion and there likely exists variability between parents' threshold to provide their child with pain medication. We could not limit their use of alternative or adjuvant agents; as such, results reflect the 'true-world' effectiveness of prescribed regimens, but not the exact effects of the medications of interest. Third, only children with an isolated acute fracture were included in this study and it is unclear whether results can be applied to other musculoskeletal injuries (e.g., sprains, strains). Fourth, there were small demographic imbalances between the two cohorts (i.e., slightly older, more fracture reductions, less buckle fractures, and more procedural sedations in the oxycodone group). This might suggest that fractures deemed clinically more important were selected into the oxycodone group. Still, this represents the reality of who received oxycodone in our clinical setting, as opioids tend to be prescribed less for younger children, and more often after reductions. These demographic imbalances should be considered when interpreting our results, in terms of applicability to local clinical populations. Finally, the Faces Pain Scale–Revised has not been formally validated for home use. We chose it as the current recommended self-report tool to assess pain levels for our study age group and has been successfully employed in the at-home setting in previous similar work [26,28].

With the over-arching goal of improving pain management, functional outcomes, and quality of life for children, further research is needed to evaluate the effectiveness of oral analgesic agents, especially in the at-home setting. Future work would benefit from utilizing a standardized approach to collection and reporting of adverse events, such as MedDRA [30]. This will allow more meaningful comparisons of adverse events across studies, as more and more, we are seeing current oral analgesic options to be reported as equipotent. Furthermore, identifying functional outcomes important to parents and children represent key patient-oriented outcome measures, ensuring future research remains relevant to children and families. Finally, future exploration of combinations of non-opioid analgesics, more potent opioids, and/or non-pharmacologic therapies is needed to determine optimal pain management strategies, particularly for more severe pain, as we have yet to identify the optimal oral agent/combination of agents to relieve orthopedic injury-related pain.

## Conclusion

A large cohort of over 300 children receiving recommendations to use either ibuprofen or oxycodone experienced similar analgesic effectiveness when treating acute fracture pain at home. Children who were prescribed oxycodone reported significantly more adverse events compared to those who received recommendations to use ibuprofen and experienced more negative impact on their functional outcomes. Children who were prescribed oxycodone used other pain medications more often than those told to use ibuprofen. Recommending that families use oxycodone to treat fracture-related pain does not appear to confer any benefit over ibuprofen for pain relief and, in fact, is associated with greater reporting of adverse events and functional limitations for the child. This study, when combined with results of previously published clinical trials, suggests that at this time, recommending ibuprofen as first-line therapy for acute fracture pain is the better option. Future clinical trials are needed to confirm these findings.

## Supporting information

**S1 File. Case report form.**
(PDF)

## Acknowledgments

Pediatric Emergency Research Canada (PERC) membership can be found at https://perc-canada.ca/pages/34-executive-committee. Current PERC executive members include Dr. Stephen B. Freedman (University of Calgary, Calgary, Alberta), Dr. Roger Zemek (University of Ottawa, Ottawa, Ontario), Dr. Samina Ali (University of Alberta, Edmonton, Alberta), Dr. Amanda S. Newton (University of Alberta, Edmonton, Alberta), Dr. Garth Meckler (University of British Columbia, Vancouver, British Columbia), Dr. Maala Bhatt (University of Ottawa, Ottawa, Ontario), Dr. Cathie-Kim Le (University of Toronto, Toronto, Ontario), Candice McGahern (University of Ottawa, Ottawa, Ontario), and Becky Emerton (University of Calgary, Calgary, Alberta). Becky Emerton is the primary contact for PERC at rebecca.emerton@ahs.ca. We thank Iram Usman for her assistance with statistical analyses. We thank the families who so kindly agreed to support our work by agreeing to participate in this study.

## Author Contributions

**Conceptualization:** Samina Ali, David W. Johnson, Rhonda J. Rosychuk, Bruce Carleton, Patrick J. McGrath, Amy L. Drendel.

**Data curation:** Samina Ali, Robin Manaloor.

**Formal analysis:** Samina Ali, Robin Manaloor, Rhonda J. Rosychuk.

**Funding acquisition:** Samina Ali, David W. Johnson, Rhonda J. Rosychuk, Sylvie LeMay, Bruce Carleton, Patrick J. McGrath, Amy L. Drendel.

**Investigation:** Samina Ali.

**Methodology:** Samina Ali, David W. Johnson, Rhonda J. Rosychuk, Sylvie LeMay, Patrick J. McGrath, Amy L. Drendel.

**Project administration:** Samina Ali.

**Resources:** Samina Ali, David W. Johnson, Rhonda J. Rosychuk, Amy L. Drendel.

**Software:** Rhonda J. Rosychuk.

**Supervision:** Samina Ali.

**Writing – original draft:** Samina Ali, Robin Manaloor.

**Writing – review & editing:** David W. Johnson, Rhonda J. Rosychuk, Sylvie LeMay, Bruce Carleton, Patrick J. McGrath, Amy L. Drendel.

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
