## [Decision Letter · Decision Letter 0]

21 Apr 2021

PONE-D-20-33147

An observational cohort study comparing ibuprofen and oxycodone in children with fractures

PLOS ONE

Dear Dr. Ali,

Thank you for submitting your manuscript to PLOS ONE. After careful consideration, we feel that it has merit but does not fully meet PLOS ONE’s publication criteria as it currently stands. Therefore, we invite you to submit a revised version of the manuscript that addresses the points raised during the review process.

Your manuscript has been assessed by two external experts whose reports are appended below. Both reviewers have raised important concerns about the study design and potential biases that may affect the ability of the results to support the conclusions - please take particular care to respond to these concerns in your revised manuscript and Response to Reviewers document. 

We look forward to receiving your revised manuscript.

Kind regards,

Dr Joseph Donlan

Senior Editor

PLOS ONE

Journal Requirements:

 [The funders had no role in study design, data collection and analysis, decision to publish, or preparation of the manuscript.].

5. One of the noted authors is a group or consortium [on behalf of Pediatric Emergency Research Canada]. In addition to naming the author group, please list the individual authors and affiliations within this group in the acknowledgments section of your manuscript. Please also indicate clearly a lead author for this group along with a contact email address.

Reviewers' comments:

Reviewer's Responses to Questions

**Comments to the Author**

1. Is the manuscript technically sound, and do the data support the conclusions?

Reviewer #1: No

Reviewer #2: Yes

2. Has the statistical analysis been performed appropriately and rigorously? 

Reviewer #1: No

Reviewer #2: Yes

3. Have the authors made all data underlying the findings in their manuscript fully available?

Reviewer #1: No

Reviewer #2: Yes

4. Is the manuscript presented in an intelligible fashion and written in standard English?

Reviewer #1: Yes

Reviewer #2: Yes

5. Review Comments to the Author

Reviewer #1: The authors report a well written paper on an interesting paper. However, the study design cannot support their conclusions. The administration of oxycodone was decided by the physician and this could lead to important bias. Moreover, the lack of blindness is a further important potential bias. An RCT would be the appropriate study design for the purposes of the authors' study. However, if the authors want to consider data from an observational study design, a propensity score analysis should be performed.

Reviewer #2: Ali and coll. made a prospective observational cohort study to compare the effectiveness and safety of ibuprofen and oxycodone for at-home management of an isolated fracture pain in 329 children aged 4-16 years (n=217 ibuprofen, n=112 oxycodone). They found that ibuprofen and oxycodone have similar analgesic effectiveness for at-home fracture pain, but oxycodone was associated with more adverse events and negatively impacted function. The study is well conducted and the topic is of interest while confirmatory of the efficacy and safety of ibuprofen for fracture-related pain.

My only concern regards the presence of a selection bias in the administration of oxycodone in children with a more severe fracture. In fact, oxycodone was more often prescribed in fractures needing procedural sedation and reduction, and in fractures later referred to the orthopedic surgeon. Also, more fractures involving the upper leg/hip, suspected for being higher energy injuries, were reported in the oxycodone group, while buckle fractures, low energy trauma, were fewer. This might suggest that fractures deemed clinically more important were selected into this group. This bias could also explain the more frequent use of adjuvant medications during the first 3 days in the oxycodone group compared to the ibuprofen group and, as a consequence, the higher adverse events incidence in children treated with oxycodone.

In my opinion, this study weakness, which is part of its observational nature, should be more highlighted and discussed.

6. PLOS authors have the option to publish the peer review history of their article (what does this mean?). If published, this will include your full peer review and any attached files.

Reviewer #1: No

Reviewer #2: **Yes: **Simone Lazzeri

---

## [Author Response · Author response to Decision Letter 0]

21 Jun 2021

Please see attached response to reviewers table.

---

## [Editor Report · Decision Letter 1]

23 Aug 2021

An observational cohort study comparing ibuprofen and oxycodone in children with fractures

PONE-D-20-33147R1

Dear Dr. Ali,

We’re pleased to inform you that your manuscript has been judged scientifically suitable for publication and will be formally accepted for publication once it meets all outstanding technical requirements.

Kind regards,

Simone Lazzeri

Guest Editor

PLOS ONE

Additional Editor Comments (optional):

None
---

## [Editor Report · Acceptance letter]

31 Aug 2021

PONE-D-20-33147R1 

An observational cohort study comparing ibuprofen and oxycodone in children with fractures 

Dear Dr. Ali:

I'm pleased to inform you that your manuscript has been deemed suitable for publication in PLOS ONE. Congratulations! Your manuscript is now with our production department. 

Kind regards, 

on behalf of

Dr. Simone Lazzeri 

Guest Editor

PLOS ONE